# Simultaneous Detection of RIG-1, MDA5, and IFIT-1 Expression Is a Convenient Tool for Evaluation of the Interferon-Mediated Response

**DOI:** 10.3390/v14102090

**Published:** 2022-09-21

**Authors:** Alexey A. Lozhkov, Marina A. Plotnikova, Marya A. Egorova, Irina L. Baranovskaya, Ekaterina A. Elpaeva, Sergey A. Klotchenko, Andrey V. Vasin

**Affiliations:** 1Institute of Biomedical Systems and Biotechnologies, Peter the Great Saint Petersburg Polytechnic University, 195251 St. Petersburg, Russia; 2Smorodintsev Research Institute of Influenza, 197376 St. Petersburg, Russia; 3Scientific and Educational Center for Biophysical Research in the Field of Pharmaceuticals, Saint Petersburg State Chemical Pharmaceutical University, 197022 St. Petersburg, Russia

**Keywords:** qPCR, cytosolic sensors, interferons, ISGs, in vitro transcribed RNAs, influenza A virus, SARS-CoV-2, SOCS-1

## Abstract

In this study, we developed a novel, multiplex qPCR assay for simultaneous detection of RIG-1, MDA5, and IFIT-1 at the mRNA level. The assay was validated in A549 cells transfected with in vitro transcribed RNAs. Both exogenous RNA-GFP and self-amplifying (saRNA-GFP) induced significant expression of RIG-1, MDA5, IFIT-1, as well as type I and III interferons. In contrast, native RNA from intact A549 cells did not upregulate expression of these genes. Next, we evaluated RIG-1, MDA5, and IFIT-1 mRNA levels in the white blood cells of patients with influenza A virus (H3N2) or SARS-CoV-2. In acute phase (about 4 days after disease onset) both viruses induced these genes expression. Clinical observations of SARS-CoV-2 typically describe a two-step disease progression, starting with a mild-to-moderate presentation followed by a secondary respiratory worsening 9 to 12 days after the first onset of symptoms. It revealed that the expression of RIG-1, MDA5, and MxA was not increased after 2 and 3 weeks from the onset the disease, while for IFIT-1 it was observed the second peak at 21 day post infection. It is well known that RIG-1, MDA5, and IFIT-1 expression is induced by the action of interferons. Due to the ability of SOCS-1 to inhibit interferon-dependent signaling, and the distinct antagonism of SARS-CoV-2 in relation to interferon-stimulated genes expression, we assessed SOCS-1 mRNA levels in white blood cells. SARS-CoV-2 patients had increased SOCS-1 expression, while the influenza-infected group did not differ from heathy donors. Moreover, SOCS-1 mRNA expression remained stably elevated during the course of the disease. It can be assumed that augmented SOCS-1 expression is one of multiple mechanisms that allow SARS-CoV-2 to escape from the interferon-mediated immune response. Our results implicate SOCS-1 involvement in the pathogenesis of SARS-CoV-2.

## 1. Introduction

Under homeostatic conditions, cells produce interferons (IFNs) at extremely low, often undetectable, levels. Induction of IFN synthesis is mediated by the activation of pattern recognition receptors (PRRs). Cytosolic sensors that belong to the category of RIG-1 like receptors (RLR) play a key role in IFN production in response to the penetration of foreign RNAs [1,2]. All RLRs have a central DExD/H-box helicase domain with ATPase activity and a carboxy-terminal domain (CTD). These domains work together and bind to foreign RNAs. The RNA sensors RIG-I and MDA5 also have two N-terminal domains termed caspase activation and recruitment domains (CARD), which are necessary for further signal transduction and immune response induction. Conversely, the third member of the RLR family, LGP-2, lacks CARD domains, and it is thought that this RNA helicase is required for regulation of RIG-I and MDA5 activity [1,2]. Cytosolic sensors of the RLR family are required for protection from RNA and DNA viruses (except ssDNA viruses) [2]. In their resting states, RIG-I and MDA5 are in a monomeric form. Binding to a foreign RNA molecule induces RNA sensor oligomerization. The activated, multimeric forms of RIG-I or MDA5 are able to interact with the MAVS protein adapter, which is anchored to the mitochondrial or peroxisome membrane [1]. The MAVS adapter activates TANK-binding kinase-1 (TBK-1) and IκB kinase-ε (IKK-ε), which leads to activation of the transcription factors IRF-3 and IRF-7. These transcription factors, together with NF-κB, induce IFN production [2].

IFNs do not have a direct antiviral effect [3]. They act in an autocrine or paracrine manner to activate JAK/STAT-dependent signaling cascades and induce the expression of hundreds of interferon-stimulated genes (ISGs). One of the most significant ISGs is IFIT-1 (ISG-56). IFIT-1 is a gene with broad-spectrum antiviral activity. The IFIT family includes IFIT-1 (ISG56), IFIT-2 (ISG-54), IFIT-3 (ISG60), and IFIT-5 (ISG58). Among them, IFIT-1 expression is the most upregulated, notably in cells that are stimulated with IFNs, dsRNAs, or viruses [4]. IFIT-1 protein has an intracellular localization. It is able to inhibit the initiation of protein synthesis at several stages. It has been shown that IFIT-1 is able to interact with eukaryotic translation initiation factor 3 subunit e (eIF3e) [4,5,6]. Constitutive IFIT-1 expression leads to a decrease in protein synthesis and cell proliferation [5]. It has been shown that IFIT-1 inhibits the replication of several RNA and DNA viruses (Lymphocytic choriomeningitis virus, West Nile virus, Parainfluenza virus, Hepatitis C virus; Human papillomavirus [6,7,8,9,10]. However, it has also been reported that IFIT proteins are involved in the induction of apoptosis. IFIT-1 gene silencing was associated with a lower level of apoptosis in A549 cells treated with 5’-PPP-ssRNA, a synthetic agonist of RIG-1 [11]. 

The aim of the study was to develop a multiplex qPCR system for simultaneous detection of MDA5, RIG-1, and IFIT-1 mRNA levels and to evaluate these genes’ expression in the white blood cells (WBC) of influenza and SARS-CoV-2 patients.

## 2. Materials and Methods

### 2.1. Cells

The A549 (CCL-185) cell line was obtained from the American Type Culture Collection (ATCC, lot #70018877, Manassas, VA, USA). A549 cells (human type II alveolar epithelial line) were cultured in F12K medium (Gibco, Grand Island, NY, USA) supplemented with 10% fetal bovine serum (Biowest, Riverside, CA, USA; South America). Treated cells and intact (mock) cells were incubated at 37 °C (5% CO_2_ with humidification). 

### 2.2. WBC Isolation

In total, the study involved 14 healthy donors, 13 patients with influenza A/H3N2 (2018–2019 epidemic season), and 17 SARS-CoV-2 pneumonia patients being treated at clinics in St. Petersburg (Russia) in 2019–2020. The following symptoms were observed as the most striking clinical manifestations in patients: fever; intoxication syndrome (weakness, headache, muscle pain); and/or catarrhal syndrome (nasal congestion, rhinorrhea, sore throat, cough, chest pain). Inclusion of patients in the A/H3N2 and SARS-CoV-2 groups was carried out based on positive results from laboratory diagnostics of the relevant pathogens in oropharyngeal swabs. Laboratory diagnosis of pathogens in selected swabs was performed by RT-PCR using certified kits. Blood for white blood cell (WBC) isolation was collected in vacuum tubes with sodium heparin. Eight milliliters of blood, diluted with DPBS to a volume of 12 mL, was introduced (avoiding mixing) into a tube containing 9 mL of Lymphosep (BioWest, Riverside, CA, USA; South America). Tubes were then centrifuged at 400× *g* for 20 min; the resulting WBC layers were taken and washed twice with DPBS containing 2% FBS. Prior to analysis, frozen cells were stored in liquid nitrogen vapor (RPMI (Gibco, Grand Island, NY, USA) storage medium containing 10% DMSO, 50% FBS). Blood samples for WBC isolation from IAV patients were collected on the 3–4 days after onset of clinical symptoms. 

For SARS-CoV-2 patients, the specific IgG antibody levels were measured. According to the results of serological examination, SARS-Cov-2 patients were divided into three groups: 3–4 days (IgG < 1); second (1 < IgG < 10) and third (IgG > 10) weeks after the onset of illness.

This clinical investigation was approved by the Institutional Ethics Committee, and the corresponding protocol numbers are 118, dated 16 October 2017 and 164, dated 12 February 2021. Informed consent was obtained from all donors and patients who provided research materials. All biological experiments were performed in accordance with the relevant guidelines and regulations.

### 2.3. In Vitro Transcription

Synthetic GFP-coding RNAs (IVT-RNA-GFP) were obtained by in vitro transcription using the mMESSAGE mMACHINE kit (Invitrogen, Carlsbad, CA, USA, #AM1344). The pJET1.2-GFP (Thermo Scientific, Waltham, MA, USA, #K1231) and T7-VEE-GFP (Addgene, Watertown, MA, #58977) plasmids, containing the T7 promoter and the GFP gene, were used as templates for RNA synthesis during in vitro transcription. One microgram of linearized plasmid template was used for each reaction. Anti-Reverse Cap Analog (ARCA) (Thermo Fisher Scientific, Waltham, MA, USA, #AM8045) was used for efficient translation of the RNA. To maximize RNA yield and the fraction of capped transcripts, we used the following cap analog/GTP ratios: 4:1 for RNA-GFP; and 10:1 for self-amplifying GFP-coding RNA (IVT-saRNA-GFP). Reactions were performed according to the manufacturer’s protocol. Following RNA synthesis, the DNA template was removed by subsequent digestion with DNase Turbo (Invitrogen, Carlsbad, CA, USA, #AM1344). The Poly(A) Tailing Kit (Invitrogen, Carlsbad, CA, USA, #AM1350) was used to polyadenylate the 3′-termini of transcribed RNA. Transcripts were purified by lithium chloride precipitation according to the recommended protocol (Invitrogen, Carlsbad, CA, USA, #AM1344). RNA samples were analyzed using electrophoretic separation under denaturing conditions. Five hundred nanograms of each RNA sample was mixed with an equal volume of Gel Loading Buffer II (Invitrogen, Carlsbad, CA, USA, #AM1344) and heated for 5 min at 80 °C. Samples were subsequently loaded into wells of 1% agarose gel (containing 0.5 µg/mL ethidium bromide) and run in 1x MOPS buffer at room temperature. 

### 2.4. Cell Transfection 

High-molecular weight (HMW, #tlrl-pic) and low-molecular weight (LMW, #tirl-picw) poly(I:C) dsRNAs, produced by InvivoGen (San Diego, CA, USA), were used as PRR agonists. Total RNA was isolated from either uninfected A549 cells or cells infected with IAV A/California/07/09 (H1N1pdm09) (24 h post infection). Total RNA was extracted from cells with TRIzol reagent (Invitrogen, Carlsbad, CA, USA) in full accordance with manufacturer’s instructions. Viral genomic RNA was isolated from influenza virus A/California/07/09 (H1N1pdm09) using the RNeasy Mini Kit (Venlo, Netherlands). RNA transfections were performed using Lipofectamine MessengerMAX Reagent (Thermo Scientific, Waltham, MA, USA) according to manufacturer instructions. There were approximately 0.6 µL of Lipofectamine Messenger Max Reagent and 1.40 × 10^11^ RNA molecules per well. For IVT RNA-GFP, total RNAs, mRNA, viral genomic RNA, LMW, and HMW, this was approximately 80, 160, 16, 40, 120, and 380 ng, respectively. The immunogenicity of the studied RNAs was evaluated 24 h after introduction of the complexes.

### 2.5. Primer and Probe Design

The target mRNA sequences of RIG-1 (DDX58), MDA5 (IFIH-1), IFIT-1, SOCS-1, AnxA-1, IFNA, IFNB, IFNL2/3, and MxA were downloaded from the GenBank database. Primers and fluorescent oligonucleotide probes, containing fluorescent reporter dyes at the 5′-end and a quencher at the 3′-end, were commercially synthesized and HPLC-purified (Evrogen, Novosibirsk, Russia).

### 2.6. RNA Isolation

Total RNA was isolated from A549 cells and WBCs using TRIZol reagent (Invitrogen, Austin, TX, USA) according to the manufacturer’s instructions. RNA concentrations were analyzed using a NanoDrop ND-1000 spectrophotometer (NanoDrop Technologies, Wilmington, DE, USA).

### 2.7. Reverse Transcription

One microgram of total RNA was treated by DNase (Promega, Madison, WI, USA) and then directly reverse transcribed using oligo-dT_16_ primers and RNAscribe RT (BioLabMix, Novosibirsk, Russia). Complementary DNA synthesis was carried out at 50 °C for 50 min. Enzyme was inactivated at 80 °C for 5 min. Products were diluted (1:2) and stored at −20 °C until use.

### 2.8. PCR Analysis

Real-time PCR assays were performed using the CFX96 Real-Time PCR System (Bio-Rad, Hercules, CA, USA). Multiplex qPCR was performed in 25 μL final reaction volumes containing 12.5 μL BioMaster HS-qPCR mix (2×) (BioLabMix, Novosibirsk, Russia) and 2 μL cDNA (obtained from 1 μg of RNA). The reaction mixtures additionally contained 200 nM of each primer and probe for RIG-1 and IFIT-1 as well as 750 nM of each primer and 250 nM of probe for MDA5. Conditions of two-step PCR were 95 °C for 5 min, followed by 40 amplification cycles (95 °C for 15 s, 61.3 °C for 30 s).

Expression of other genes was assessed by monoplex two-step qPCR (500 nM of each primer and 200 nM of probe). IFNA expression was measured using HS-qPCR SYBR Blue (2×) (BioLabMix, Novosibirsk, Russia). 

### 2.9. ELISA

Human IL-8 concentrations in cell culture supernatants were measured by enzyme-linked immunosorbent assay (ELISA) using the IL-8/CXCL-8 DuoSet ELISA commercial kit (DY208, R&D Systems, Minneapolis, MN, USA).

### 2.10. Statistical Data Processing

Data processing was carried out in Microsoft Excel. GraphPad Prism was used to evaluate the statistical significance of differences.

## 3. Results

### 3.1. Multiplex Optimization

Multiplex qPCR is one of the most accurate and sensitive methods for simultaneous evaluation of gene expression [12]. However, this method requires intensive optimization as primer mismatches and imperfect qPCR conditions may lead to the amplification of non-specific products. We developed a qPCR set for simultaneous detection of cytosolic RNA sensors (RIG-1, MDA5) as well as IFIT-1 (a key antiviral ISG) at the mRNA level.

The analyzed genes are localized on different chromosomes. The lengths of their mature transcripts range from 600 to 4700 nucleotides. Regarding mRNA transcription variants: the RIG-1 gene (DDX58) features seven; the IFIT-1 gene features five; and the IFIH-1 gene (MDA5) features one (*Home*—*Nucleotide*—*NCBI (nih.gov)*). The designed sequences (forward and reverse primers, TaqMan probe) should detect almost all of the transcriptional variants of the target genes both in monoplex and multiplex assay (Appendix A). Sequence selection was carried out based on: a desired melting temperature for primers of 60 °C and above; and an oligonucleotide probe melting temperature over 63 °C. Sequences designed to detect human RIG-1, IFIT-1, and MDA5 mRNA were analyzed in BLAST (http://blast.ncbi.nlm.nih.gov/, accessed on 15 June 2022) to identify their possible range of nonspecific products. Sequences highly homologous to any other human genes were not found (Appendix A). Specific amplification products are shown in Appendix A.

RNAs isolated from intact cells and infected cells (influenza A virus) were used to make cDNA matrix for qPCR optimization. Primer concentrations and thermal conditions were optimized to amplify all of the specific PCR products with high efficiency. The amplification efficiencies (***E***) of the target and reference genes must be almost equal. The E value for each amplified gene was determined by plotting a graph with the *x*-axis as log(quantity) and the *y*-axis as Ct obtained in monoplex and multiplex qPCR (Figure 1). The efficiency of reaction was estimated using the equation: ***E* = 10^(–1/slope)^ − 1**. To achieve highly efficient amplification of specific fragments for all three genes in multiplex qPCR, primer and probe concentrations ranged from 50 to 1000 nM. The optimal primer and probe concentrations, determined for each gene, are presented in the Materials and Methods section. The corresponding multiplex PCR amplification E values were: RIG-1 (98%); MDA5 (98%); and IFIT-1 (97%).

### 3.2. IFN-Mediated Response to Transfection

Cytosolic RNA sensors are activated in response to viral penetration into the cytoplasm. Synthetic or viral RNAs are perfect activators of RIG-1 or MDA5, which induce IFN and ISG expression. Therefore, using multiplex qPCR, we first estimated gene expression in response to stimulation of A549 cells with: synthetic agonists of TLR (HMW and LMW poly(I:C) RNAs); IVT-RNA-GFP; IVT-saRNA-GFP; or native RNAs (total RNA, mRNA, miRNA isolated from intact A549 cells). Transfection of A549 cells with IVT-RNA-GFP and IVT-saRNA-GFP resulted in stable translation of green fluorescent protein after 24 h (Appendix A).

Both IVT-RNA-GFP and IVT-saRNA-GFP led to an increase in IFN mRNA synthesis. In addition, the magnitude of changes in IFNA mRNA levels was much less than IFNB or IFNL2/3 (Appendix A). It was also revealed that IVT-RNA-GFP and IVT-saRNA-GFP are even more potent inductors of IFNB or IFNL2/3 mRNA than poly(I:C) dsRNAs. SOCS-1 expression was also increased upon transfection of cells with synthetic RNAs (Appendix A). To elucidate whether activation of IFN mRNA expression leads to further activation of the immune response, we measured IL-8 production. The results show that all of the stimulations by synthetic RNAs induced robust production of IL-8, with poly(I:C) RNAs inducing the largest secretion (Appendix A).

Next, using the developed multiplex qPCR, we evaluated RIG-1, MDA5, and IFIT-1 levels in response to transfection. Stimulation with synthetic RNAs led to a strong increase in the expression of these genes in A549 cells (Figure 2). It is noteworthy that we did not detect significant changes in IFNA, IFNL2/3 or ISG expression in response to the introduction of native RNAs (mock, cellular) into the cell. Since upregulated RIG-1, MDA5, IFIT-1 and IFN expression was detected only in response to stimulation of A549 cells with synthetic RNAs, we evaluated the levels of these genes in cells stimulated by viral RNAs: isolated from influenza virions (viral RNA); or isolated from influenza infected cells (IAV-infected total RNA). 

It was shown that both total RNA from infected cells and viral RNA are able to dramatically induce RIG-1, MDA5, and IFIT-1 expression. Total RNA from intact A549 cells did not influence these ISGs or IFNs at the mRNA level (Figure 3). Taking into account that total RNAs from intact and influenza-infected cells were isolated according to the outlined method in a parallel manner, we conclude that total RNA from intact host cells did not stimulate an IFN-mediated immune response.

### 3.3. Multiplex Testing with WBCs

For practical evaluation of the developed multiplex qPCR, we carried out an experiment with human WBCs. Cells were isolated from the whole blood of healthy volunteers or respiratory illness patients (influenza A/H3N2 or SARS-CoV-2). ISGs analysis was carried out in the acute phase of illness (3–4 days from onset of initial symptoms). We used Glyceraldehyde-3-phosphate dehydrogenase (GAPDH) and Actin beta (ACTB) as reference genes (Appendix A). In acute phase both viruses induced the expression of all analyzed genes (RIG-1, MDA5, and IFIT-1) (Figure 4a–c).

We investigated possible correlation between IFIT-1 and RLR expression in WBCs in both infected groups and in healthy volunteers (Appendix A). There was a strong, positive correlation between expression of IFIT-1 and RLR only in the SARS-CoV-2 group (r2 = 0.91 for RIG-1 and r2 = 0.93 for MDA5) at the acute phase of the illness. Nonetheless, the correlation coefficient in influenza infected patients was a bit lower. These results indicate that there is a relationship between the RNA-sensors and IFIT-1 expression in the early stages of viral infection. Interestingly, IFIT-1 is considered a negative regulator of RIG-1 or MDA5 cytosolic RNA helicase signaling [13]. Additional studies are required for more accurate calculation and a deeper comprehension of the relations.

A canonical antiviral ISG, MxA, which is considered a reliable marker of viral infection [14], was elevated in both IAV and SARS-CoV-2 patients (Appendix A).

Due to the ability of SOCS-1 to inhibit IFN-dependent signaling [15,16,17], and the distinct antagonism of SARS-CoV-2 in relation to IFNs and ISGs, we decided to assess the level of SOCS-1 mRNA in WBCs. Interestingly, SARS-CoV-2 patients had increased SOCS-1 expression, while the influenza-infected group did not differ from healthy donors (Figure 4d).

Next, to assess the formation of a systemic immune response in patients with SARS-CoV-2, the ISGs expression study was also performed in dynamics at different phases of the disease (Figure 5).

Interestingly, expression of RIG-1, MDA5, and MxA was remarkably increased compared to the control in WBC of SARS-CoV-2 patients only at early stage of the disease (Figure 5a,b,d). In contrast, SOCS-1 mRNA expression levels remained stably elevated during all phases of illness (Figure 5e). We also detected that IFIT-1 expression was significantly increased only at 4 and 21 days after the first onset of symptoms (Figure 5c). These data indicate that the robust expression of SOCS-1 in SARS-CoV-2 patients is associated with impaired IFN signaling and lowered ISGs expression. An increased level of IFIT-1 can be considered as a mechanism for suppressing both protein synthesis and virus replication [18]. Otherwise, augmented IFIT-1 expression could be associated with apoptosis [11].

## 4. Discussion

The RNA helicases RIG-1 and MDA5 are key cytosolic sensors that induce type I and type III IFN expression. Simultaneous analysis of their mRNA levels enables assessment of IFN-mediated immune response activation. The third target gene for the developed multiplex is IFIT-1, a canonical ISG with distinct antiviral activity. It should be noted that IFN mRNA levels in WBCs can be extremely low or even undetectable. In our work, it was shown that the developed multiplex qPCR is suitable for RIG-1, MDA5, and IFIT-1 mRNA measurement in A549 cells and WBCs.

A promising technology is vaccine development based on synthetic RNA molecules [19]. The immune response induced by the penetration of foreign RNA is one of the possible limitations of this approach. We used our PCR system to evaluate the cellular response to transfection with various native and synthetic RNAs, including IVT RNAs functionally similar to RNA vaccines. After the introduction of IVT RNAs, an increase in the expression of PRRs (RIG-1, MDA5) was noted. It has been shown that penetration of exogenous RNA into the cell is associated with endosomal TLR and cytosolic sensor activation, IFN and ISG expression, as well as proinflammatory cytokine and chemokine production [20].

Synthetic poly(I:C) dsRNA and long viral dsRNA are considered to be MDA5 agonists; uncapped ssRNA with a 5′-triphosphate (5′-PPP) or 5′-diphosphate (5′-PP) group activates RIG-1 [1]. The presence of 5′-PPP (or 5′-PP) in mRNA is not the only condition for RIG-1 activation. The simultaneous presence of a 5′-PPP or 5′-PP group, a duplex ‘panhandle’ structure, and the absence of a 2′-O-methyl group in the 5′-terminal nucleotide are characteristic of viral RNA (i.e., influenza A RNAs), but not cellular cytosolic RNA [2]. Thus, IVT RNAs formed during transcription, contaminated with uncapped RNA and containing 5′-PPP groups, can act as agonists of cytosolic RNA sensors and stimulate type I and III IFNs, as well as ISG expression.

Inhibition of translation is associated with the recognition of foreign RNA by IFIT-1. IFIT-1 attenuates translation either at the level of 43S complex stabilization, or at the level of RNA binding, interacting through a 2′-O-unmethylated 5′ cap structure or 5′-PPP group [3,9,10,21]. The latter mechanism seems to be more selective. 

Therefore, the developed multiplex qPCR assay is suitable for assessing IFN-dependent response activation when cells are transfected with exogenous RNAs. This is especially useful for evaluating the immunogenicity of mRNA-based drugs. Estimation of IFIT-1 mRNA level is not only an indicator of ISG expression, but also indicates the degree of protein synthesis inhibition. Increases in the mRNA levels of RIG-1, MDA5, and IFIT-1 were correlated with augmented expression of IFNB, IFNL2/3, as well as SOCS-1. Interestingly, the IVT RNA, despite 5′-capping and 3′-polyadenylation, turned out to be an even more potent inducer of IFNs than control poly(I:C) dsRNA. These data once again emphasize the need to use modified bases and thorough purification of mRNA [20].

Interferons exhibit distinct antiviral activity against SARS-CoV-2 [22,23,24]. Severe cases can occur early in the disease course, but clinical observations typically describe a two-step disease progression, starting with a mild-to-moderate presentation followed by a secondary respiratory worsening 9 to 12 days after the first onset of symptoms [25]. Despite virus induced activation of immune cells, overproduction of pro-inflammatory cytokines and chemokines (TNFα, IL-6, IL-8, CXCLs), and the possible development of acute respiratory distress syndrome (ARDS), a key feature of the infection is low production of type I and type III IFNs in the early stages of illness [26,27]. Moreover, in WBCs isolated from the whole blood of SARS-CoV-2 patients, it has been shown that the immune-related expressed genes pattern depends on the severity of infection. Patients with moderate severity were characterized by increased ISG levels. In critical patients, however, extremely weak activation of antiviral ISGs (MxA, IFITM, IFIT) was noted, and IFN-α/β production was significantly reduced at the protein level [25]. Thus, patients with mild disease display a coordinated pattern of ISG expression across the WBC population; immune ISG-expressing cells are systemically absent in patients with severe disease [28]. Consequently, IFN production in the initial phase of SARS-CoV-2 infection is a key factor determining the course of illness [27]. A cytokine storm is associated with severe outcome of viral respiratory infections. SARS-CoV-2 patients have a higher serum level of cytokines (TNF-α, IFN-γ, IL-2, IL-4, IL-6 and IL-10) and C-reactive protein (CRP) [29]. The level of IL-6 and IL-10 is especially increased in critical patients [25,29]. In previous work [30], we have also discovered increased serum level of IL-10, IL-6, as well as acute-phase proteins (CRP and Serum amyloid A1) in the same patients with SARS-CoV-2 induced viral pneumonia. This indicates that the investigated group of patients with SARS-CoV-2 infection suffered from exuberant systematic inflammation.

Activation of cytosolic RNA-sensors leads to IFN induction, while IFIT-1 can be considered an antiviral ISG that blocks the primary translation of (+) RNA viruses. At the early stage of the disease we have discovered that RIG-1, MDA5, IFIT-1, MxA, and SOCS-1 expression was elevated. High ISG expression at the acute phase of disease has been accompanied by low SARS-CoV-2 antibody levels (IgG < 1). These data can be considered as a common innate antiviral response. For instance, influenza infected patients had augmented mRNA level of the genes, with the exception of SOCS-1.

Considering the complex dynamics of RNA sensing with the consequent dysregulation of IFN production, we compared ISG expression in WBCs of SARS-CoV-2 patients at approximately 14 and 21 days after disease onset. Our experiments demonstrated that only SOCS1 expression was significantly increased during 4, 14 and 21 days from onset of initial symptoms. At the same time, expression of other ISGs (RIG-1, MDA5, IFIT-1, and MxA) at 14 and 21 days did not differ from the control group. In another research it was also demonstrated that at about ten days after disease manifestation, severe SARS-CoV-2 patients were characterized by downregulation of key antiviral ISGs [25]. Our results are in line with this observation. The increase in SOCS-1 expression is especially notable as it is well known that SARS-CoV-2 can suppress ISGs expression [25] and SOCS-1 is an endogenous inhibitor of IFN action [15].

SARS-CoV-2 utilizes many mechanisms to escape the immune response [27]. It simultaneously suppresses the IFN-mediated immune response at several levels. This suggests: avoidance of virus recognition by RNA helicases; suppression of IFN production by inhibiting RIG-I/MDA5-dependent signaling pathways, and inhibition of JAK/STAT-dependent signaling cascades. SARS-CoV-2 M protein inhibits RIG-1 activity, which negatively affects the production of IFN and ISGs (IFIT-1, ISG-15) [31]. At the level of IFN signaling, M protein promotes autophagic degradation of STAT1. Degradation of this transcription factor also suppresses IFIT-1 expression [31]. In addition to M protein, at least four SARS-CoV-2 proteins (NSP5, ORF7a, N, ORF6) suppress the activity of JAK/STAT-dependent signaling cascades [27].

It is well known that SOCS proteins negatively regulate IFN signaling by inhibiting the JAK-STAT signaling pathway [15]. It has been shown that influenza A virus can induce the expression of SOCS-1, which inhibits STAT1 phosphorylation. Moreover, JAK-STAT signaling disruption by increased SOCS-1 protein results in the activation of NF-κB, leading to excessive IFN-λ production with impaired antiviral response [17]. It emerged that SOCS-1 expression was significantly increased in patients with SARS-CoV-2, but not in those infected with influenza A virus. It can be assumed that the augmented SOCS-1 expression is one of multiple mechanisms that allow SARS-CoV-2 to escape from the IFN-mediated immune response. In recent research [16], it was demonstrated that SARS-CoV-2 accessory protein ORF3a dampens IFNs signaling via upregulation of SOCS-1. Higher SOCS-1 expression in COVID-19 patients correlates with SARS-CoV-2’s greater pathogenicity compared to the influenza A virus. A more detailed study of SOCS features during COVID-19 is required. Our results argue in favor of SOCS-1 involvement in SARS-CoV-2 pathogenesis and the possibility of using this gene as one of the markers of coronavirus infection.

## 5. Conclusions

We developed a multiplex qPCR assay for simultaneous evaluation of RIG-1, MDA5, and IFIT-1 expression. Using the assay, we discovered that RIG-1, MDA5 and IFIT-1 mRNA levels were augmented in the WBCs of influenza infected patients, but not in those from SARS-CoV-2 patients after 2 weeks from the disease onset. This motivated us to assess SOCS-1 expression. It was revealed that SARS-CoV-2 patients had increased SOCS-1 expression during the course of the disease, while the influenza-infected group did not differ from healthy donors.

## Figures and Tables

**Figure 1 viruses-14-02090-f001:**
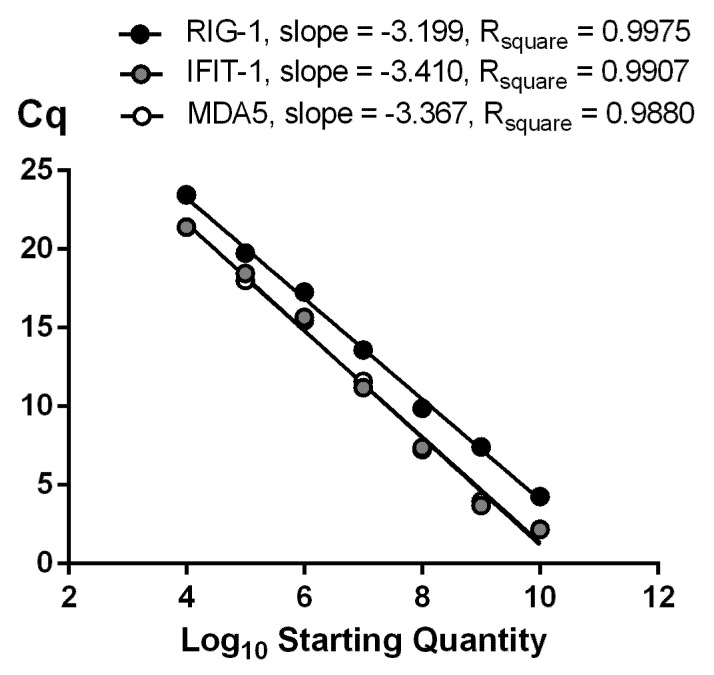
**Determination of amplification efficiencies of RIG-1, MDA5, and IFIT-1 genes in multiplex qPCR****.** Cq cycles versus log starting quantity were plotted to calculate the slopes. The corresponding qPCR efficiencies were calculated according to the equation ***E* = 10^(−1/slope)^ − 1**.

**Figure 2 viruses-14-02090-f002:**
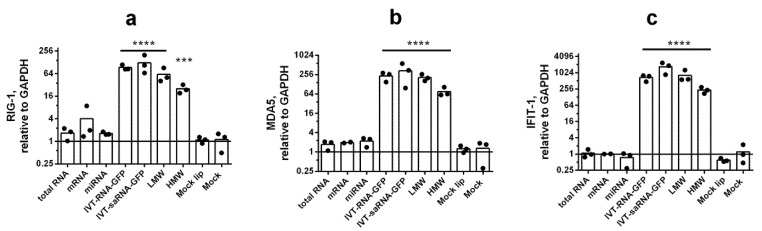
**Synthetic RNAs are powerful activators of the IFN-mediated response.** The expression levels of RIG-1 (**a**), MDA5 (**b**), and IFIT-1 (**c**) were detected at 24 h post transfection. Gene expression was analyzed via the ΔΔCt method (relative to GAPDH). Statistical significance (*p*-value) was determined by ordinary one-way ANOVA, followed by a pairwise Holm–Sidak’s multiple comparisons test: ****—adjusted *p* Value < 0.0001; ***—<0.001 compared to Mock. Mock—intact cells that were cultured in the same conditions and were not transfected (instead, sterile medium F12K was added). At least three biological replicates were used for each experimental data point. Data are represented as median.

**Figure 3 viruses-14-02090-f003:**
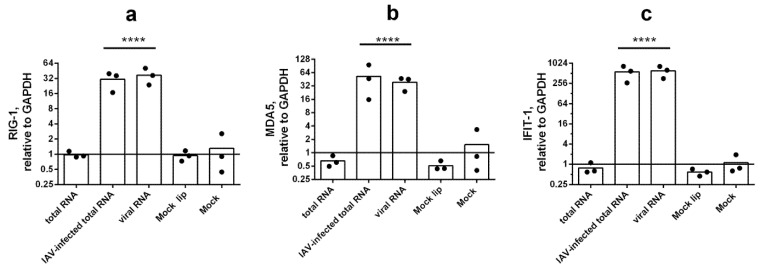
**Total RNA from IAV infected cells strongly induces RIG-1 (a), MDA5 (b), and IFIT-1 (c) expression.** Expression levels were detected at 24 h post transfection. Gene expression was analyzed via the ΔΔCt method (relative to GAPDH). Statistical significance (*p*-value) was determined by ordinary one-way ANOVA, followed by Holm–Sidak’s multiple comparisons test: ****—adjusted *p* Value < 0.0001 compared to Mock. Mock—intact cells that were cultured in the same conditions and were not transfected (instead, sterile medium F12K was added). At least three biological replicates were used for each experimental data point. Data are represented as median.

**Figure 4 viruses-14-02090-f004:**
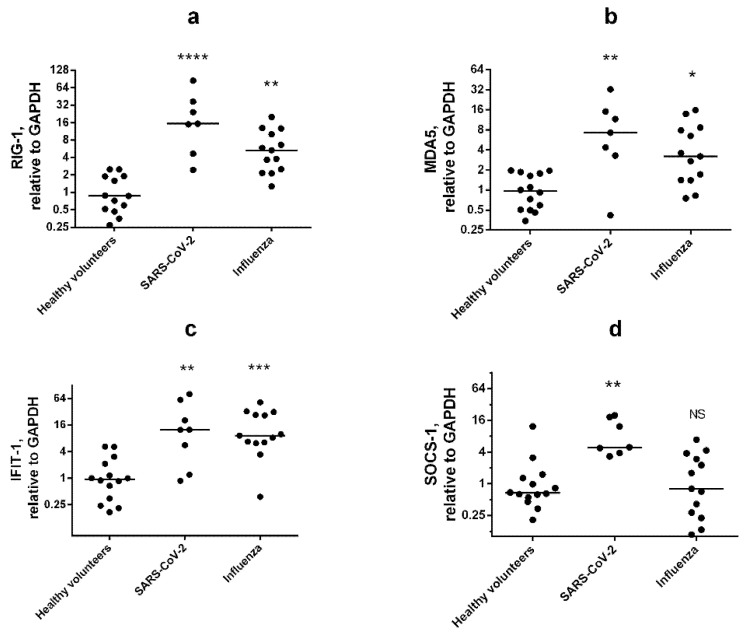
**IAV and SARS-Cov-2 are potent inducers of RIG-1, MDA5, and IFIT-1 expression in WBCs.** Gene expression of RIG-1 (**a**), MDA5 (**b**), IFIT-1 (**c**), and SOCS-1 (**d**) was analyzed via the ΔΔCt method (relative to GAPDH). Statistical significance (*p*-value) was determined by Kruskal–Wallis test, followed by Dunn’s multiple comparisons test; NS—no significant difference; ****—adjusted *p* Value < 0.0001; ***—< 0.001; **—<0.01; *—<0.05 compared to healthy volunteers. Data are represented as median.

**Figure 5 viruses-14-02090-f005:**
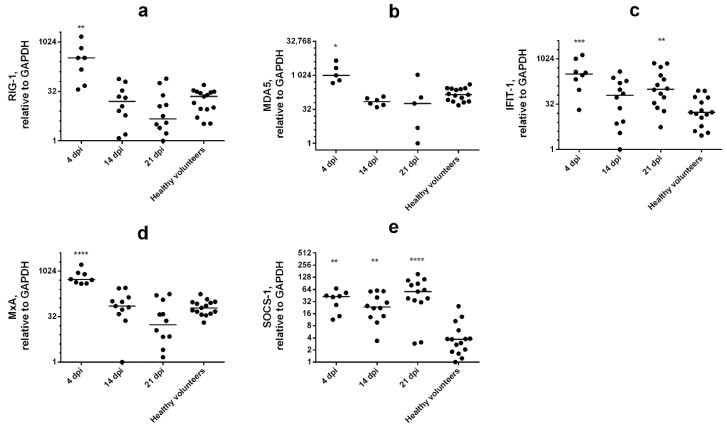
**SOCS-1 expression is increased in WBCs during SARS-CoV-2 infection.** The mRNA levels of RIG-1 (**a**), MDA5 (**b**), IFIT-1 (**c**), MxA (**d**), and SOCS-1 (**e**) were measured via the ΔΔCt method (relative to GAPDH). Statistical significance (*p*-value) was determined by Kruskal–Wallis test, followed by Dunn’s multiple comparisons test; ****—adjusted *p* Value < 0.0001; ***—<0.001; **—<0.01; *—<0.05 compared to healthy volunteers. Data are represented as median.

## Data Availability

All data generated and analyzed during this study are included in this article.

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
