# Peer review of "Simultaneous Detection of RIG-1, MDA5, and IFIT-1 Expression Is a Convenient Tool for Evaluation of the Interferon-Mediated Response"

_viruses, 2022, doi:10.3390/v14102090_

Round 1
Reviewer 1 Report
The manuscript by Lozhkov et al. reports a qPCR method to detect simultaneously the expression of 3 genes (RIG-I, MDA5, IFIT-1) that are induced by viral infection and exposure of cells to foreign RNA. Then, they use this qPCR to detect these 3 genes in RNA samples obtained from Covid and flu patients. The first part of the study, i.e. concerning the method, is satisfactory with some minor modifications (see below), particularly in assessing the interferon response when cells are transfected with exogenous RNAs. The second part is more problematic than the first one. First of all, as also indicated by the authors, the number of patients is low, but more importantly, the analysis was carried out with RNA collected in a single time point after infection. Knowing the complex dynamics of RNA sensing with the consequent dysregulation of interferon production, this second part requires additional analysis with longitudinal samples including the SOCS-1 expression that is one of the multiple mechanisms that allow SARS-CoV-2 to escape from interferon.
Minor points:
line 68: report the aim of the study
line 84: define better the clinical characteristics of the patients. Catarrhal syndrome is not satisfactory.
line 85: swab. Do you mean nasal, oropharyngeal swab?
line 149: indicate the quantity of the cDNA used rather than the volume
ACTB and GAPDH: report the full name the first time they appear in the text
line 260: what do you mean for nuances of SARS-CoV-2 therapy?
Author Response
Comments and Suggestions for Authors
The manuscript by Lozhkov et al. reports a qPCR method to detect simultaneously the expression of 3 genes (RIG-I, MDA5, IFIT-1) that are induced by viral infection and exposure of cells to foreign RNA. Then, they use this qPCR to detect these 3 genes in RNA samples obtained from Covid and flu patients. The first part of the study, i.e. concerning the method, is satisfactory with some minor modifications (see below), particularly in assessing the interferon response when cells are transfected with exogenous RNAs. The second part is more problematic than the first one. First of all, as also indicated by the authors, the number of patients is low, but more importantly, the analysis was carried out with RNA collected in a single time point after infection. Knowing the complex dynamics of RNA sensing with the consequent dysregulation of interferon production, this second part requires additional analysis with longitudinal samples including the SOCS-1 expression that is one of the multiple mechanisms that allow SARS-CoV-2 to escape from interferon.
Response: Thank you for this comment. Initially, the manuscript was supposed to devote only to a methodological study describing the design of qPCR and its validation using human cell cultures, as well as biomaterials isolated from humans. As samples to assess the sensitivity of qPCR, we used samples from patients with influenza and SARS-CoV-2. The results we obtained seemed interesting to us and we decided to speculate on this topic in the Discussion section. In response to your comment, we have performed and included in the manuscript additional qPCR results (for RIG-I, MDA5, IFIT-1, and SOCS-1 genes) for SARS-CoV-2 patients in different illness stage. Corresponding changes have been made to the Materials and Methods, Results, Discussion:
We added the information to the section 2.2. WBC isolation:
… In total, the study involved 14 healthy donors, 13 patients with influenza A/H3N2 (2018-2019 epidemic season), and 17 SARS-CoV-2 pneumonia patients being treated at clinics in St. Petersburg (Russia) in 2019-2020. The following symptoms were observed as the most striking clinical manifestations in patients: fever; intoxication syndrome (weakness, headache, muscle pain); and/or catarrhal syndrome (nasal congestion, rhinorrhea, sore throat, cough, chest pain). Inclusion of patients in the A/H3N2 and SARS-CoV-2 groups was carried out based on positive results from laboratory diagnostics of the relevant pathogens in oropharyngeal swabs …
… Blood for white blood cell (WBC) isolation was collected in vacuum tubes with sodium heparin …
… Blood samples for WBC isolation from IAV patients were collected on the 3-4 days after onset of clinical symptoms. According to the results of serological examination, SARS-Cov-2 patients were divided into three groups: 3-4 days (IgG < 1); second (1 < IgG < 10) and third (IgG > 10) weeks after the onset of illness …
We added the information to the section 3.3. Multiplex testing with WBCs:
…A canonical antiviral ISG, MxA, which is considered a reliable marker of viral infection (Haller, 2015), was elevated in both IAV and SARS-CoV-2 patients (Supplementary Figure S8)…
…Next, to assess the formation of a systemic immune response in patients with SARS-CoV-2, the ISGs expression study was also performed in dynamics at different phase of disease (Figure 5). Interestingly, expression of Rig-1, Mda-1, and MxA was remarkably increased compared to the control in WBC of SARS-CoV-2 patients only at early stage illness (Figure 5 a,b,d). In contrast, SOCS-1 mRNA expression levels remained stably elevated during all phases of illness (Figure 5e). We also detected IFITM-1 expression was significantly increased only at 4 21 day after the first onset of symptoms (Figure 5c). These data indicate that the robust expression of SOCS-1 in SARS-CoV-2 patients is associated with impaired IFN signaling and lowered ISGs expression. An increased level of IFIT-1 can be considered as a mechanism for suppressing both protein synthesis and virus replication (Pichlmayr, 2011). Otherwise, augmented IFIT-1 expression could be associated with apoptosis (Yap, 2020)…
Changes have been made to Figure 4:
Figure 4. IAV is potent inducer of MDA5 and IFIT-1 expression in WBCs. Gene expression of RIG-1 (a), MDA-5 (b), and IFIT-1 (c) was analyzed via the ΔΔCt method (relative to GAPDH). Statistical significance (p-value) was determined by Kruskal-Wallis test, followed by Dunn’s multiple comparisons test; *** ― adjusted P Value < 0.001; ***― < 0.01; * ― < 0.05 compared to healthy volunteers. Data are represented as median.
We added the new Figure 5 in manuscript:
Figure 5. SOCS-1 expression is increased in WBCs during SARS-CoV-2 infection. The mRNA levels of RIG-1 (a), MDA5 (b), IFIT-1 (c), MxA (d), and SOCS-1 (e) were measured via the ΔΔCt method (relative to GAPDH). Statistical significance (p-value) was determined by Kruskal-Wallis test, followed by Dunn’s multiple comparisons test; **** ― adjusted P Value < 0.0001; *** ― < 0.001; **― < 0.01; * ― < 0.05 compared to healthy volunteers. Data are represented as median.
We added the information to the section Discussion:
… Interferons exhibit distinct antiviral activity against SARS-CoV-2 (Mantlo, 2020; Plotnikova, 2021; Hoagland, 2021). Severe cases can occur early in the disease course, but clinical observations typically describe a two-step disease progression, starting with a mild-to-moderate presentation followed by a secondary respiratory worsening 9 to 12 days after the first onset of symptoms (Hadjadj, 2020). Despite virus induced activation of immune cells, overproduction of pro-inflammatory cytokines and chemokines (TNFα, IL-6, IL-8, CXCLs), and the possible development of acute respiratory distress syndrome (ARDS), a key feature of the infection is low production of type I and type III IFNs in the early stages of illness (Blanco-Melo, 2020; Zhang, 2021). …
… Cytokine storm is associated with severe outcome of viral respiratory infections. SARS-CoV-2 patients have higher serum level of cytokines (TNF-α, IFN-γ, IL-2, IL-4, IL-6 and IL-10) and C-reactive protein (CRP) (Han, 2020). The level of IL-6 and IL-10 is especially increased in critical patients (Han, 2020; Hadjadj, 2020). In previous work (Taraskin, 2022) we also have discovered increased serum level of IL-10, IL-6, as well as acute-phase proteins (CRP and Serum amyloid A1) in patients with SARS-CoV-2 induced viral pneumonia. This indicates that the considered group of patients with SARS-CoV-2 infection was suffered from exuberant systematic inflammation. …
… Activation of cytosolic RNA-sensors leads to IFN induction, while IFIT-1 can be considered an antiviral ISG that blocks the primary translation of (+) RNA viruses. At the early stage of the disease we have discovered that RIG-1, MDA5, IFIT-1, MxA, and SOCS-1 expression was elevated. High ISG expression at acute phase of disease has been accompanied by low SARS-CoV-2 antibody levels (IgG <1). These data can be considered as a common innate antiviral response. For instance, influenza infected patients had augmented mRNA level of the genes, with the exception of SOCS-1. …
Minor points:
line 68: report the aim of the study
Response: The aim of the study has been included in the text of the manuscript:
- Intoduction: …The aim of the study was to develop a multiplex qPCR system for simultaneous detection of MDA5, RIG-1, IFIT-1 mRNA level and to evaluate these genes expression in white blood cells (WBC) of influenza and SARS-CoV-2 patients….
line 84: define better the clinical characteristics of the patients. Catarrhal syndrome is not satisfactory.
Response: Thanks for this clarification. Our study involved patients with a laboratory-confirmed (RT-PCR) diagnosis of influenza (2018-2019 epidemic season) and SARS-Cov-2 (2020) being treated in specialized hospitals in St. Petersburg (Russia). Patient information card included the following parameters: fever, presence of severe or mild pneumonia, artificial ventilation, days of hospitalization, cycle threshold values in RT-PCR, days from the onset of the disease, serological test, peripheral oxygen saturation, C-reactive protein (CRP). The information is now reflected below and in Materials and methods of the manuscript (presented in the answer to the question earlier).
line 85: swab. Do you mean nasal, oropharyngeal swab?
Response: A more accurate formulation, reflecting the methodology has been included in the text of the manuscript:
2.2. WBC isolation: ….Inclusion of patients in the A/H3N2 and SARS-CoV-2 groups was carried out based on the presence of a catarrhal syndrome and positive results from laboratory diagnostics of the relevant pathogens in oropharyngeal swabs. Laboratory diagnosis of pathogens in selected swabs was performed by RT-PCR using certified kits.
line 149: indicate the quantity of the cDNA used rather than the volume
Response: A more accurate formulation, reflecting the methodology has been included in the text of the manuscript:
2.8. PCR analysis: …Multiplex qPCR was performed in 25 μL final reaction volumes containing 12.5 μL BioMaster HS-qPCR mix (2x) (BioLabMix, Russia) and 2 μL cDNA (obtained from 1 μg of RNA).
ACTB and GAPDH: report the full name the first time they appear in the text Fixed in text
Response: The corresponding gene names has been included in the text at the first mention:
3.3. Multiplex testing with WBCs: …We used Glyceraldehyde-3-phosphate dehydrogenase (GAPDH) and Actin beta (ACTB) as reference genes (Supplementary Figure S5)....
line 260: what do you mean for nuances of SARS-CoV-2 therapy?
Response: Thank you. This wording was not quite accurate; we deleted it in the manuscript. The study was conducted on samples obtained from patients at the very beginning of the global pandemic. At the first stage of the pandemic, there were significant difficulties associated with obtaining false-negative PCR results with very convincing clinical and radiological signs of pneumonia. At the same time, symptomatic therapy had been starting even before laboratory confirmation of the disease, and this could affect the parameters we evaluated (ISGs expression). In addition, in our case, this might led to an incorrect definition of the day of the disease.
In the revised manuscript, we analyzed moderate to severe SARS-Cov-2 patients, who were divided into three groups: the early phase of the disease, during the first 3-4 days after the onset of symptoms, the phase of average duration (2 weeks after the first onset of symptoms) and the recovery stage (3 weeks after the onset of symptoms). Comparison of SARS-Cov-2 and IAV patients was carried out in the early phase of the disease.
3.3. Multiplex testing with WBCs: …Perhaps this is due to differences in the course of the disease…

Reviewer 2 Report
In this paper, Lozhkhov et al describes the development of a multiplexed qPCR assay which can detect key genes that play a role in IFN responses to viral infection. Further to this, they use their assay to measure IFN responses in white blood cells of patients infected with Influenza and SARS-COV-2. The development of the qPCR assay it methodologically sound however the manuscript can be improved by addressing some of the following points:
· What was the rationale for selection of IFIT proteins as opposed to other common IFN stimulated genes such as ISG15, IFTMs or Mx?
· Some further explanation for why IL 8 levels were measured would make the manuscript clearer
· .Figure S4- description says IL 8 expression levels have a similar expression pattern to ISGs, which is not accurate. IL8 secretion is highest with HMW poly IC whereas IFN induction and ISG expression is highest with IVT-SARNA-GFP.
· Supplementary figure S6 and S7 are not discussed at all the manuscript. Why where they included? It would be useful to know why IFIT-1 expression levels had higher correlation with RIG-I than MDA 5.
· The conclusion statement “The assay may be useful for screening RNA-based antivirals for 383 innate immune response ability.” Isn’t very well supported by the evidence presented so far. Perhaps more discussion around this would be useful.
Minor comments.
Line 58: Are IFIT proteins cytosolic or associated with any intracellular organelle?
Line 63: Please give examples of some DNA and RNA viruses inhibited by IFIT 1
Author Response
Comments and Suggestions for Authors
In this paper, Lozhkhov et al describes the development of a multiplexed qPCR assay which can detect key genes that play a role in IFN responses to viral infection. Further to this, they use their assay to measure IFN responses in white blood cells of patients infected with Influenza and SARS-COV-2. The development of the qPCR assay it methodologically sound however the manuscript can be improved by addressing some of the following points:
What was the rationale for selection of IFIT proteins as opposed to other common IFN stimulated genes such as ISG15, IFTMs or Mx?
Response: Thank you for the question. It is well known that RIG-1 and MDA5 are RNA-sensors that are able to detect synthetic or viral RNA (reviewed in Rehwinkel, 2020). In turns, it was shown that IFIT proteins can interact with 2'-O-unmethylated 5' cap structure or 5'-triphosphate group (PPP-RNA) of a mRNA molecule, thereby acting as a sensor of viral single-stranded RNAs and inhibiting expression of viral messenger RNAs (Daffis, 2010; Kimura, 2013; Li, 2015; Rabbani, 2016). IFIT-1 binds PPP-RNA with nanomolar affinity. In the absence of IFIT-1, the growth and pathogenicity of viruses containing PPP-RNA is much greater (Pichlmair, 2011). This way, RIG-1,MDA5, and IFIT-1 form a unique system of cell’s protection from foreign RNAs.
Moreover, there is some evidence that IFIT proteins are involved in IFNs-induced apoptosis. So that, one can speculate that a regulatory axis (RIG-1/MDA5-IFNs-IFIT) exists (Yap, 2020). What is more, our results (part 3.3 Multiplex testing with WBCs) indicate correlation between RNA-sensors and IFIT-1 expression in infected patients, especially in the case of RIG-1 in COVID-19 patients.
Some further explanation for why IL 8 levels were measured would make the manuscript clearer
Response: Thank you for your question. Type I IFNs and pro-inflammatory cytokines (TNF-α, IL-1β, IL-6, IL-18 etc) (Guo, Thomas, 2017) are presented in cell culture supernatants at low levels. In control A549 cells they could be even undetectable. In the same time, IL-8 level presents in amounts of about dozen pg/ml in cell control and is extremely elevated in response to synthetic agonists (such as LMW, HMW, IVT-RNA-GFP). IL-8 is required the recruitment of neutrophils and other immune cells to the site of infection (Bickel, 1993) and is elevated in influenza infected A549 cells (as it is shown in a figure below). This way, measurement of IL-8 level also allows eliciting A549 cell’s immune response activation in case of RNA transfection.
Figure: Measurement of secreted IL-8 level in A549 cells supernatants 24 hpi with various respiratory viruses (our unpublished data). The y-axis shows IL-8 concentration in pg/ml; the x-axis shows viruses used with cells. IBV – Influenza B virus, B/Phuket/3073/13) (Yamagata lineage); H1 – Influenza A virus, A/California/07/09 ((A)H1N1pdm09); H3 – Influenza A virus, A/Texas/50/12 ((A)H3N2); RSV – Respiratory syncytial virus, strain A2; AdV1 – Adenovirus type 5 (strain Adenoid 75). The statistical significance of differences were determined by the Kruskal-Wallis and Dunn’s multiple comparisons test. A group of intact cells was taken as the control group; * - P <0.05
Figure S4 - description says IL 8 expression levels have a similar expression pattern to ISGs, which is not accurate. IL8 secretion is highest with HMW poly IC whereas IFN induction and ISG expression is highest with IVT-SARNA-GFP.
Response: We thank for the accurate wording. Changes have been included in the supplementary file:
Figure S4. IL-8 secretion is elevated in response to exogenous RNA
Supplementary figure S6 and S7 are not discussed at all the manuscript. Why where they included? It would be useful to know why IFIT-1 expression levels had higher correlation with RIG-I than MDA 5.
Response: We thank for the remark. Changes have been included in the manuscript (part 3.3 Multiplex testing with WBCs):
…We investigated possible correlation between IFIT-1 and RLR expression in WBCs in both infected groups and in healthy volunteers (Supplementary Figure S6 and S7). There was a strong, positive correlation between expression of IFIT-1 and RLR only in the SARS-CoV-2 group (r2 = 0.91 for RIG-1 and r2 = 0.93 for MDA5) at the acute phase of the illness. Nonetheless, the correlation coefficient in influenza infected patients was a bit lower. These results indicate that there is a relationship between the RNA-sensors and IFIT-1 expression in the early stages of viral infection. Interestingly, IFIT-1 is considered a negative regulator of RIG-1 or MDA5 cytosolic RNA helicase signaling (Li, 2009). Additional studies are required for more accurate calculation and a deeper comprehension of the relations…
The conclusion statement “The assay may be useful for screening RNA-based antivirals for innate immune response ability.” Isn’t very well supported by the evidence presented so far. Perhaps more discussion around this would be useful.
Response: Thank you for the remark. We have deleted the sentence. We further plan to include additional multiplex qPCR sets for MxA, 2'-5'-oligoadenylate synthetase 1 (OAS-1), Protein kinase R (PKR), and Interferon-induced transmembrane protein 1 (IFITM-1) genes for more complex evaluation of cells immune response to synthetic or viral RNAs.
Minor comments.
Line 58: Are IFIT proteins cytosolic or associated with any intracellular organelle?
Response: Thank you for the comment. IFIT-1 mainly is localized in cytosol. Reference: (https://www.proteinatlas.org/ENSG00000185745-IFIT1/subcellular).
Line 63: Please give examples of some DNA and RNA viruses inhibited by IFIT 1.
Response: Thank you for the comment. We added the information to the manuscript (part 1 Introduction):
…It has been shown that IFIT-1 inhibits the replication of several RNA (Lymphocytic choriomeningitis virus, West Nile virus, Parainfluenza virus, Hepatitis C virus) and DNA (Human papillomavirus) viruses (Wacher, 2007; Raychoudhuri, 2011; Saikia, 2010; Kimura, 2013; Rabbani, 2016)…

Round 2
Reviewer 1 Report
The manuscript is significantly improved.
In Fig. 4 panel d, ** are misplaced .
Author Response
Response: We thank for the remark. We have made changes in the Figure 4(d) and its caption.
